# In-layer inhomogeneity of molecular dynamics in quasi-liquid layers of ice
Ikki Yasuda [ID], Katsuhiro Endo, Noriyoshi Arai [ID] & Kenji Yasuoka [ID] ✉

Quasi-liquid layers (QLLs) are present on the surface of ice and play a significant role in its distinctive chemical and physical properties. These layers exhibit considerable heterogeneity across different scales ranging from nanometers to millimeters. Although the formation of partially ice-like structures has been proposed, the molecular-level understanding of this heterogeneity remains unclear. Here, we examined the heterogeneity of molecular dynamics on QLLs based on molecular dynamics simulations and machine learning analysis of the simulation data. We demonstrated that the molecular dynamics of QLLs do not comprise a mixture of solid- and liquid water molecules. Rather, molecules having similar behaviors form dynamical domains that are associated with the dynamical heterogeneity of supercooled water. Nonetheless, molecules in the domains frequently switch their dynamical state. Furthermore, while there is no observable characteristic domain size, the long-range ordering strongly depends on the temperature and crystal face. Instead of a mixture of static solid- and liquid-like regions, our results indicate the presence of heterogeneous molecular dynamics in QLLs, which offers molecular-level insights into the surface properties of ice.

Ice develops quasi-liquid layers (QLLs) on its surface in a process known as ice premelting. The QLLs endow the ice surface with unique chemical and physical properties that have important implications for the climate and environment. For example, QLLs facilitate ion formation in clouds, leading to the generation of thunderstorm via collisions. They also reduce surface friction to promote glacial movement[1,2]. Because the formation of QLLs contributes to the slipperiness of ice[3–6], they are crucial for related engineering applications such as anti-slip automobile tires and anti-freezing coatings and additives. More recently, the catalytic properties of QLLs have opened up their potential use as nanomaterial reactors[7–10].

QLLs arise when the temperature exceeds at least 20 K below the melting point[11–15]. According to experimental and simulation studies, their thickness ranges from a few molecular layers (approximately two layers or 0.9 nm) up to 10 nm[16–18]. Sanchez et al. reported the QLL layers melt by layers, where the second molecular layer from the interface melts at around 260 K[19]. Moreover, hexagonal ice is the most common ice crystal structure found in nature. This structure has three distinct crystal faces: the basal (0001) face, bilayer face of the primary prism (1010), and a monolayer structure found at the secondary prism ($\bar{1}2\bar{1}0$) faces, as shown in Fig. 1a. The surfaces of these faces lack the pairs of molecules necessary to form hydrogen bonds, resulting in the disordered molecular structures of QLLs. The properties of QLLss such as the surface thickness and diffusion coefficient vary among the three faces[20,21].

Recently, there has been an increasing awareness of the heterogeneous nature of QLLs, demanding a deeper understanding of their structural scale and dynamics[18]. Sazaki et al. experimentally investigated the appearance of droplet and sheet phases with a thickness of one molecular layer, and these structures were reported to span a width of nanometers to millimeters[22,23]. The structures are metastable formations that arise by the different types of wetting during the process of vapor deposition and evaporation[24]. They are created at just a few Kelvins below the melting temperature on top of the comparatively flat QLLs[15], whereas the underlying QLLs are also hypothesized to possess nanoscale heterogeneity. Specifically, the QLLs comprise nanometer-scale sections of solid- and liquid-like configurations, i.e., a partially melted structure[25]. It has been demonstrated that the QLLs possess a rough interface that is entropically favored, and the roughness is highly temperature-dependent[26,27]. Also, the ice-like regions serve as substrates for crystal growth[28]. In those studies, the liquid- or solid-like nature of molecules was identified based on their surrounding environments. The ice structure was identified via different techniques and order parameters[29,30], as well as through the application of machine learning (ML) models[31–33]. Despite the detection of ice crystalline structures, these formations gradually emerge as the temperature increases, rather than undergoing a sudden change in the molecular structure and thermodynamic properties in a first-order transition[34]. This implies that the nonstructural properties of molecules in the QLLs, such as their dynamic properties, may differ from those of ice crystals. Water at the vapor interface exhibits distinctive molecular

Department of Mechanical Engineering, Keio University, Yokohama, Japan. ✉e-mail: yasuoka@mech.keio.ac.jp

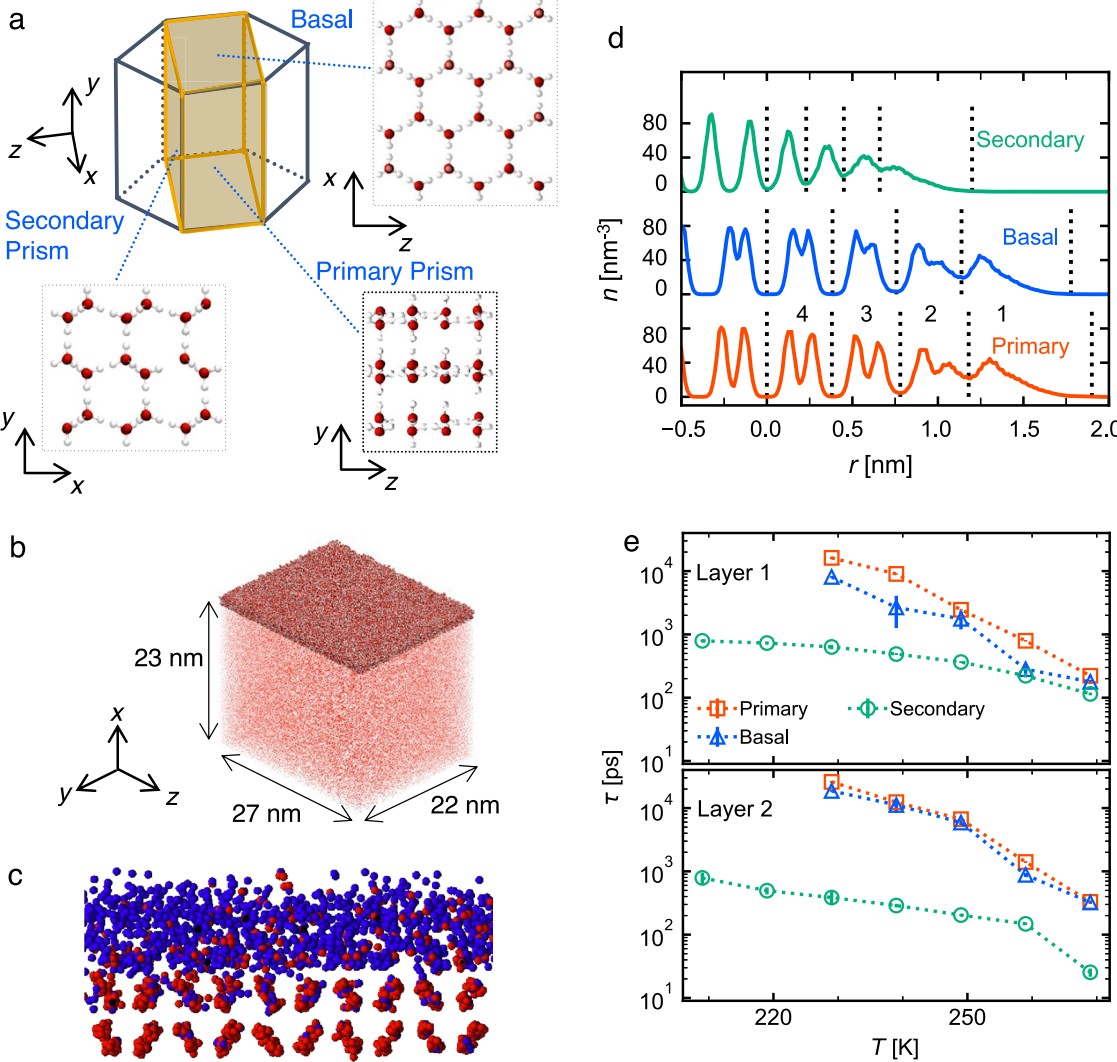

**Fig. 1 | Molecular dynamics (MD) simulation of quasi-liquid layers (QLLs). a** The primary prism, basal, and secondary prism faces of hexagonal ice in their top view. Oxygen and hydrogen atoms are denoted by red and white balls, respectively. **b** A representative snapshot of the system containing four quasi-liquid layers (QLLs) on the primary prism face at 269 K. Quasi-liquid layers are shown in darker colors and additional water molecules underneath are in lighter colors. **c** Close-up image of the QLLs. For the sake of clarity, only oxygen atoms are shown. The red and blue colors denote molecules in solid- and liquid-like dynamical states, respectively. **d** Number density $n$ profiles of oxygen atoms along the normal direction to the ice-vapor interface. The density profiles of the three faces at 269 K are illustrated. Relative position $r$ is measured from the bottom of Layer 4. **e** Relaxation time of layer transition. Survival probability function $C(t)$ is fitted into exponential curve $C(t) = C_0 \exp(-t/\tau)$ where $\tau$ is the relaxation time (see Methods for details). The standard deviation are calculated for the mean of time sections for every 5 ns and for both interfaces of the slab.

properties such as solid-like structural ordering[35] and enhanced dynamics that differs from the bulk state depending on the distance from the interface, where the structures are not necessarily coupled to the dynamics[36]. Because QLLs behave similarly to the liquids, the surface effect manifests in both the molecular dynamics and nanoscale clusters. For instance, Kling et al. showed a higher diffusion coefficient in QLLs compared to that in the bulk and identified the molecular diffusion as glass-like subdiffusion[21]. However, their simulations were limited in size and did not reveal changes in the dynamics in solid- and liquid-like clusters. This highlights the need to elucidate the molecular dynamics of QLLs with solid- and liquid-like segments at the nanoscale. The simulation cell size has substantial effects on the observed dynamics. For instance, bulk water forms immobile and mobile regions, and the immobile region becomes the nucleus of ice coupled with local ice tetrahedral structures[37]. Supercooled liquids typically display dynamical heterogeneity[38–40], which is likely linked to the molecular dynamics of QLLs.

To demonstrate the dynamical heterogeneity of the QLLs, this study performed extensive molecular dynamics (MD) simulations using premelting systems of large sizes. First, an ML approach was developed to evaluate the molecular dynamics of water by differentiating between immobile and mobile molecules that correspond to solid- and liquid-like molecules, respectively. Generally, the necessity of applying the ML approach lies because of the validity of data-driven ML approaches against complex MD data[41,42]. Next, we reveal the dynamical behavior of QLLs and demonstrate spatially biased distributions of mobile molecules in the QLLs. Finally, to clarify the nature of the dynamical clusters, we reveal the time required for molecules to remain in the dynamic liquid- and solid-like states, and the studied long-range ordering of the domains.

## Results

### Structure and dynamics of QLLs

We performed all-atom MD simulations of the ice/vapor interface system in a large simulation cell using the TIP4P/Ice model[43] with different faces of ice

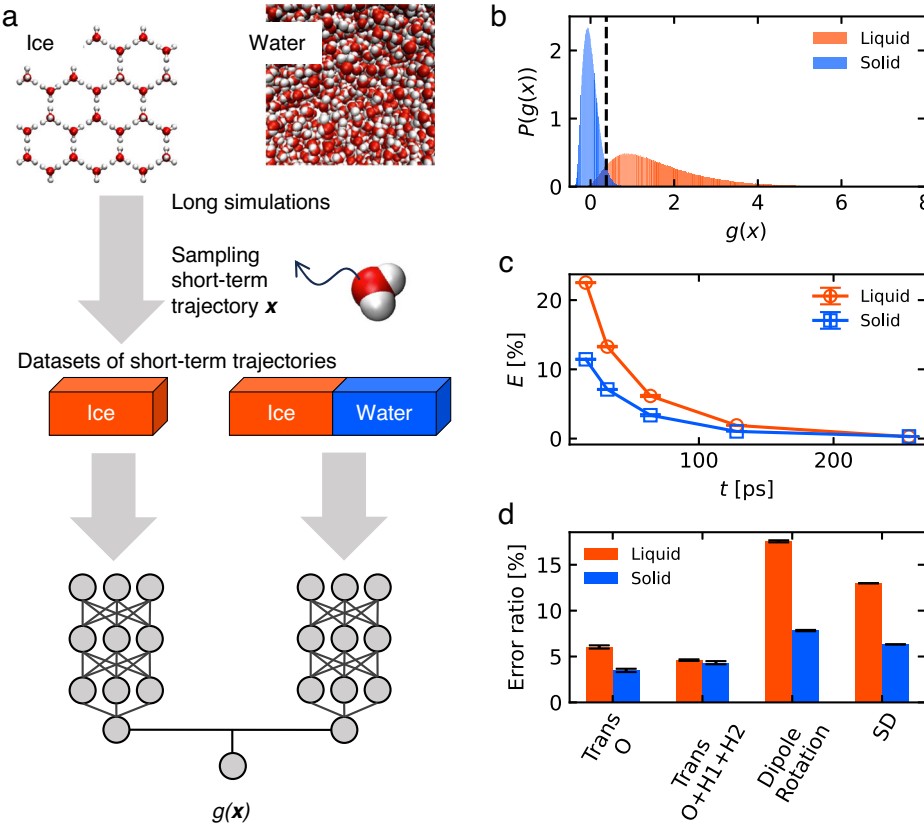

**Fig. 2 | Classification of the dynamics of liquid- and solid-like water molecules using a machine learning approach. a** Overview of the machine learning approach. Two datasets, one for solid and one for the solid-and-liquid combined system, were created from MD simulations of bulk systems. The letter means individual simulation data of the solid and liquid systems are combined. The short-term trajectories *x* are sampled from the long simulations, and input to a deep neural network that is trained via unsupervised learning. The output is a scalar $g(\mathbf{x})$. Classification of solid/liquid was performed based on $g(\mathbf{x})$. **b** Probability density distribution $P(g(\mathbf{x}))$ of the bulk liquid and solid water that are contained in the combined system. The black dashed line shows the boundary to classify solid and liquid. The displacement of oxygen atom over 64 ps was used as the input. **c** Error ratio $E$ at different trajectory lengths. Red: ratio of misclassifying liquid as solid, blue: ratio of misclassifying solid as liquid. **d** Error ratio when using different types of trajectory data as input: displacements of the oxygen atom (Trans O), displacements of oxygen and hydrogen atoms (Trans O+H1+H2), displacements of dipole moment (Dipole rotation), and square displacements (SD). In (**c**, **d**), error bars are estimated by three individual ML calculations using the same MD dataset.

(Fig. 1a). In the slab simulation, QLLs appeared spontaneously (Fig. 1b, c), and the simulations were continued for 150 ns for convergence with the last 50 ns used for analysis. The QLLs were clearly identified based on the disordered density distributions (Fig. 1d). We classified the top four-layers from the interface as QLLs and denoted them as Layers 1–4 from the topmost to the bottommost. The total thickness of the four layers was approximately 1.8 nm for bilayer faces and 1.2 nm for the secondary prism face at 269 K, which mostly unchanged in the range 209–269 K.

The layered structure of the QLLs causes distinct in-plane and interplane molecular diffusions. In-plane dynamics determines the short-term fluidity of the layers, such that in-lane diffusion of the top-most layer is glass-like but more diffusive than that of supercooled water[21]. In contrast, the interplane molecular diffusions related to longer-term behavior after the relaxation time of residence within an identical layer. Interlayer diffusion allows molecules in Layers 3-4 to move up to Layers 1-2, diffuse within the upper layers, and hop down to the original layer[44]. Here, we focus on the in-plane dynamics, and the time scale of the interlayer transition should be estimated to determine an effective time scale to investigate in-layer dynamics. For that purpose, we calculated the survival time correlation to stay in the identical layers for Layer 1 and 2 (see Methods for details and Supplementary Fig. 1 for the autocorrelation functions). At 269 K, which is close to the melting point of the model, the survival probability relaxes in the order of hundreds of ps (Fig. 1e). The layer transition was faster in the secondary prism faces than in the bilayer faces. The transition between layers occurred much more frequently in the secondary prism face than in the primary prism and basal faces, probably because the sparse monolayer structure of the secondary prism allowed a high degree of molecular fluctuations. In the following sections, we use short-term trajectories (64 ps) to distinguish the liquid- and solid-like states. Since this duration is shorter than the time scale of interlayer transition in most of the systems studied here, it is reasonable to analyze the dynamics for each layer.

## Liquid/Solid classification based on short-term molecular dynamics

In the analysis of water molecules on interfaces in simulations using TIP4P/Ice models, local structures are commonly classified to be solid or liquid based on the order parameters[27,45]. Recently, machine learning approaches to detect local ice-like structures have been developed[30–32,46]. However, as the connection between structure and dynamics is not supposed, we introduce a method to distinguish the phase of a single molecule based on the dynamics. The dynamic properties of a single water molecule include its rotational[47] and translational motions[36,37]. One of the brute force analyses in dynamics would be based on square displacements (SD). There are two notable points to be addressed (1) It is difficult to distinguish the dynamics of supercooled water at least based on the SD, and (2) high-time resolution classification is required due to frequent layer transition. Here, we developed an ML scheme to classify the state of a single molecule as either solid or liquid based on its short-term dynamics (Fig. 2a). Essentially, this ML analysis carries out a dimensional reduction to measure the dynamical differences from ice molecules. In the following sections, we first test this method in bulk solid and liquid systems and then apply it to the QLLs based on the bulk results.

We created two types of data from the MD trajectories of the bulk solid and bulk liquid systems. The first dataset contains MD trajectories of the solid system, and the second dataset contains combined trajectories, that is, the MD trajectories derived from individual simulations of both of the solid and liquid. These datasets were compared to each other using unsupervised learning. For the short-term trajectory of a single molecule *x*, the scalar value $g(\mathbf{x})$[48] that describes the statistical difference of the trajectory from the trajectories bulk ice was calculated,

$$g(\mathbf{x}) = \mathbb{E}_{\mathbf{x}' \sim \mathbf{y}'}\left[f^*(\mathbf{x}) - f^*(\mathbf{x}')\right] \qquad (1)$$

where *x* and *x*′ are short-term trajectories from the combined data and solid data, respectively. **y**′ is the probability distribution of **x**′, and $\mathbb{E}_{\mathbf{x}' \sim \mathbf{y}'}$ means taking the average of mini-batch of the solid data. The variable *x* can represent different dynamical data such as molecular translation, rotation,

or their combinations, which are available from the time-series generated by MD simulation. The function $f^*$ is the function expressed by a deep learning model, optimized to calculate Wasserstein distance between the two datasets (see Methods and Supplementary Methods for details)[48,49].

For the bulk systems of solid and liquid water at 269 K, their short-term trajectories were evaluated using $g(x)$ for the translational dynamics of oxygen atom over 64 ps (Fig. 2b). Because the distributions of $g(x)$ were well separated, we define $\alpha = 0.3775$ as the threshold separating the solid and liquid states. Yet, the distributions of solid and liquid water overlap slightly, indicating that some motions are common to both states. The liquid molecule shows a large variation in $g(x)$, corresponding to a higher degree of freedom of motion that is in contrast to the persistent bonds formed in ice. The accuracy increased at the cost of time resolution (Fig. 2c). For the application to QLLs, the time length of the trajectory needs to be set by considering the timescales of both interlayer transitions and changes in mobility within the layer. At an input time length of 64 ps, the classification errors were lower than 10% for both liquid and solid molecules. Since the variance of accuracy among individual ML runs is minor, the classification error originates from the intrinsic nature of the MD system. When using a shorter period 32 ps, there is supposed to be more confusion between solid and liquid molecules. This would not be beneficial to clarify the transition of the two states. When using a longer period of 128 ps, this would lead to confusion of layers, i.e., the properties of Layer 1 and 2 would be similar. Although analysis using a longer period could be practical by removing the molecules that transit layers, it results in the removal of large parts of molecules and thus we do not emply this approach.

Next, to select the type of dynamics used for classification, we compared the distribution of $g(x)$ obtained using different dynamical variables as inputs, namely the diffusion of oxygen and hydrogen atoms and the dipole rotation (Fig. 2d and Supplementary Fig. 2). By considering both hydrogen atoms, rotational movements in the three axes can also be considered, but the accuracy only increases slightly compared to the results using only the oxygen atom. Also, the dipole vector $d$ is defined as $d = r_O - r_M$, where $r_O$ and $r_M$ are the positions of oxygen atom and dummy atom of the same molecule, respectively. The displacements of dipole with time were significantly ambiguous in dictating the state of single molecules within a short time (Fig. 2d and Supplementary Fig. 2). Moreover, the ML method, which can consider the time series of displacements, was more accurate than the classification based on squared displacements (Fig. 2d and Supplementary Fig. 2) that only considered the displacement after a certain time step. Considering computational efficiency and time resolution, we used the dynamics of the oxygen atom over 64 ps in the following analysis. We use one of the trained models and the threshold value of 0.3775, where the combination of the model and threshold value produces the error for liquid-like molecules as 6.21% and for solid-like molecules as 3.57%.

Compared with the structure-based approach, the dynamics-based approach has some advantages. First, it enables single molecule detection because the interactions with neighbors are not calculated. Second, the computational cost is lower than that of other order parameters such as Schimit order parameters, which require calculations for pairwise atoms. The independence from neighboring atoms may be particularly suitable for surface structures that could differ from both the bulk solid and bulk liquid. Nevertheless, the dynamics-based approach is difficult to apply at low temperatures, because the dynamics is significantly suppressed and almost identical to that of the liquid.

## Dynamical heterogeneity of QLLs

The dynamics-based classification of liquid- and solid-like molecules was validated in the bulk at 269 K. Subsequently, we applied this approach to molecules in QLLs of the three faces at 269 K. Using the deep neural network trained for the bulk systems, the profile of $g(x)$ was calculated for each of Layers 1-4. According to the profile of $g(x)$, the dynamics changes according to the distance from the interface (Fig. 3a–c). For all faces, the topmost Layer 1 showed a higher $g(x)$ than that of the liquid. This implies that Layer 1 was more diffusive than bulk supercooled water[21]. Layer 2 was intermediate

between the solid and liquid, and Layers 3 and 4 were almost solid-like. For Layers 2-4, the secondary prism face showed higher $g(x)$ than the primary prism and basal faces, which is probably because of the thinner layer than the bilayer structures. Figure 3d show trajectories with low and high $g(x)$ values, in order to clarify the relationship between $g(x)$ and molecular mobility. The ice-like trajectories with low $g(x)$ display cage-trapping behavior, and the liquid-like trajectories with high $g(x)$ are random walks. In other words, a low $g(x)$ value indicates ice-like molecules with low mobility, whereas a high $g(x)$ indicates liquid-like molecules with high mobility. The ML method automatically detects differences between these two types of dynamics. Cage-trapping and jumping dynamics coexist in the QLLs.

The dynamics differs significantly between water and ice molecules over a sufficiently long period of time, and the two states in the bulk show different peaks of $g(x)$. If the QLL consists of molecules that remain in the ice and liquid states, then the distribution of $g(x)$ should display two peaks. However, our results for Layers 1-4 failed to show such distinct peaks corresponding to the solid and liquid states. There are two possible explanations for this: either the QLLs behave differently from both the solid and liquid states, or that most molecules in the QLLs frequently switch between the two states. For layer 1, the first case can be caused by enhanced molecular dynamics near the interface[36]. In particular, a higher but glassy diffusion was reported for the first layer of QLLs[21].

Supercooled water exhibits dynamical heterogeneity in the bulk, as well as other supercooled liquids. In the QLLs, ice-like clusters are present, in addition to the intrinsic heterogeneity of the supercooled water. These factors are presumed to bring in-plane dynamical heterogeneity in QLLs, not only the dynamic changes that depend on the distance from the surface. To clarify this point, we investigated the spatial distribution of $g(x)$ by coloring molecules according to their $g(x)$ values. The snapshots show the heterogeneous distribution of $g(x)$ (Fig. 4a–c). For all three faces, Layer 1 consisted of liquid-like mobile parts, solid-like inmobile parts, and voids. Most of the parts were liquid-like, whereas a small fraction of the solid part showed crystalline patterns in the bilayer structures (Fig. 4a, b). These three parts appear as randomly arranged clusters. In the secondary prism face, Layer 1 contained larger voids compared to the other two faces (Fig. 4c). In Layer 2, mobile and immobile regions are significantly separated. Based on these observations, we speculate that QLLs consist of voids, liquid-like parts, and solid-like parts, and the three parts enhance the diversity of molecular dynamics in QLLs. For example, the presence of the voids, which can be associated with the surface roughness, allows molecules in Layer 1 to leave the liquid-like parts and to move over long distances via entering the vapor parts. In addition, subdiffusion may occur because the void region reduces the accessible areas for the liquid. On the other hand, the partial crystalline structure facilitates the formation of ice-like structures that span between Layers 1–3.

A quantitative comparison of the dynamical heterogeneity between supercooled water and ice is of great interest. In addition, because the three-dimensional coordinates of the oxygen atom were used as input for ML, moving in the normal direction to the surface may have affected the results. To clearly demonstrate the in-plane dynamical heterogeneity, we used the non-Gaussian parameter $\alpha_2(t)$ to quantify the degree of dynamical heterogeneity[50] (see Methods). Glass shows short-term subdiffusion within cages ($\beta$ relaxation) and long-term diffusion caused by cage rearrangement ($\alpha$ relaxation)[51]. While $\alpha_2(\Delta t)$ is zero for homogeneous diffusion, heterogeneous dynamics is characterized by the increase and decrease of $\alpha_2(\Delta t)$, showing $\beta$ relaxation and $\alpha$ relaxation, respectively[38,52].

For QLLs, trajectories of molecules that stay in the identical layer for $\Delta t$ was analyzed. As the temperature decreases from 269 to 209 K, $\alpha_2(\Delta t)$ of Layer 1 increases (Fig. 4d), similar to the behavior of bulk supercooled water[53]. This indicates that Layer 1 is liquid-like. On the other hand, Layer 2 has the highest $\alpha_2(t)$ at 259 K (Fig. 4d). This trend of the dynamical heterogeneity can be interpreted as the effect of confinement is significant until a certain point between 259 and 269 K, and further heating alleviates the confinement effect. Then, the heterogeneity among layers and faces are compared. Just below the melting point (269 K), the dependence of Layer 1

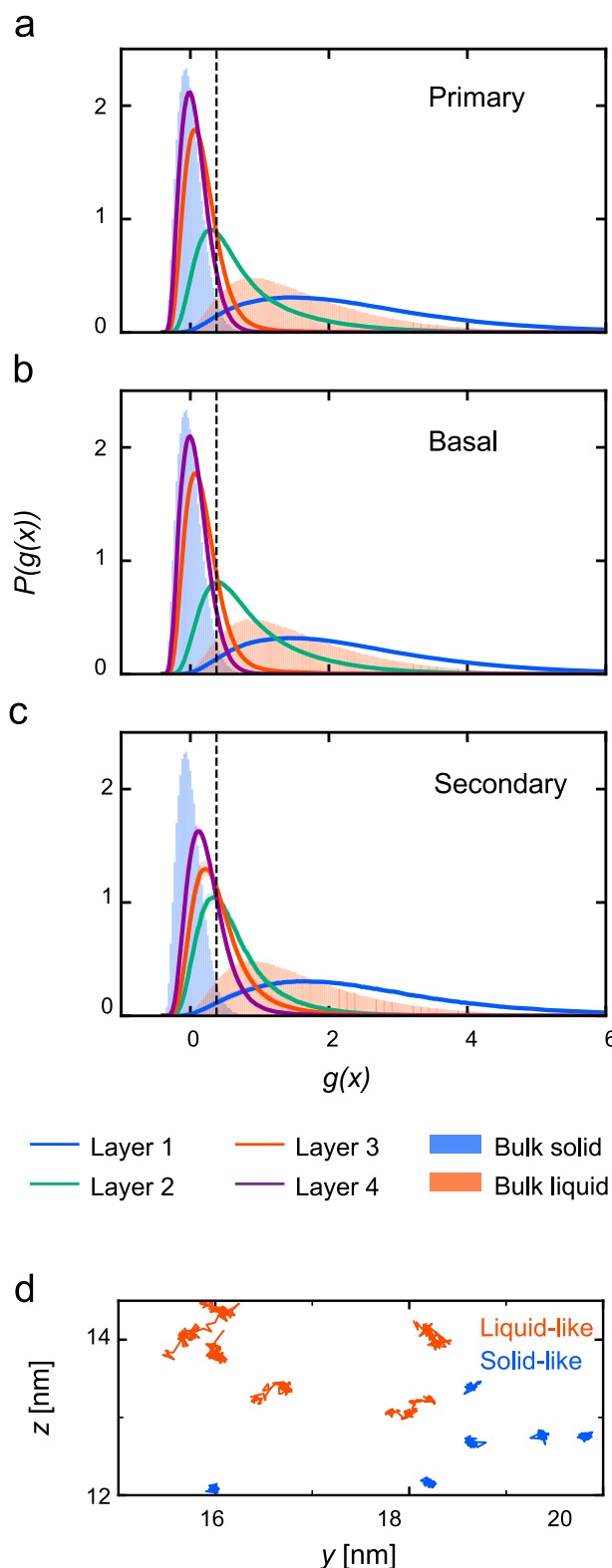

**Fig. 3 | Analysis of dynamics of oxygen molecules in QLLs at 269 K.** Profile of $g(x)$ at 269 K for Layer 1-4 of (**a**) primary prism, (**b**) basal, and (**c**) secondary prism faces, where $P(g(x))$ is the probability density. The profiles of bulk liquid and solid at 269 K are also shown, together with the threshold value for solid-liquid classification (black dashed vertical line). **d** Representative short-term trajectories over 64 ps indicated by high and low $g(x)$, with red denotes $3 \leq g(x)$ and blue denotes $g(x) \leq 0$. The trajectories were extracted from Layer 2 of the primary prism face at 269 K.

on the faces almost disappears (Fig. 4e); however, their heterogeneity is higher than that of supercooled water. Despite the limitation of the sampled data, Layer 2 demonstrates dynamical heterogeneity that relaxes slower than the supercooled liquid and may be similar to the solid state. The primary prism shows the highest degree of heterogeneity. At a low temperature of 209 K, dynamical heterogeneity occurs in Layer 1, with a peak similar to that of supercooled water but at much higher heterogeneity (Fig. 4e). The heterogeneity is the highest on the basal face.

**Liquid-like molecular ratio at different faces and temperature**

Properties of QLLs such as the thickness, diffusion coefficient, and roughness vary upon cooling. We have observed temperature-dependent changes in the dynamical heterogeneity, which would contribute to the surface nature of ice. Yet, the origin of the heterogeneity is not clarified. Herein, by evaluating the ratio of the liquid-like molecules based on the $g(x)$ from the ML method, we compare the surface nature among different faces at varying temperature. Simulations and analysis of $g(x)$ at the different faces and temperatures were performed. The threshold value of $g(x)$ at 269 K in the bulk was established and employed to calculate the liquid-like ratio of the premelting systems at different temperatures. In other words, we consider the liquid at 269 K to be the "ideal" liquid and use it as a reference to detect liquid-like molecules at a lower temperature. As a note, the necessity of this approach comes from the fact that most regions of liquids are immobile at very low temperatures, and the threshold becomes ambiguous.

The ratios of the liquid-like molecules in Layers 1 and 2 of the three faces are shown in Fig. 5a, while Supplementary Fig. 3 shows the corresponding $g(x)$ distributions. In Layer 1, the primary prism face has the lowest liquid-like ratio among the three faces at lower than 249 K. Above this temperature, all faces showed a similar ratio. In Layer 2, the liquid-like ratio was largely equal at temperatures lower than 249 K. Between 249 and 259 K, the basal faces became more fluidic, and face-dependent variance appeared. Layers 3 and 4 were composed of solid-like molecules (Supplementary Fig. 4). Based on these observations, the face dependency appears at lower than 239 K in Layer 1 and higher than 239 K in Layer 2, and it decreases as approaching the melting point. This trend of face dependency is consistent with the analysis of dynamical heterogeneity.

Although we classified the molecular dynamics into the solid- or liquid-like state, this treatment needs caution considering the QLLs behavior is not identical to the bulk in terms of the $g(x)$ profile. There are two possible reasons for the unique dynamics of QLLs from the perspective of single-molecule dynamics. Either the dynamics is unique unlike those in either solid or liquid states, or the molecules frequently switch between the two states. To determine which reason is more likely, we evaluated the survival time correlation of the liquid- and solid-like states (Fig. 5b). For comparison among different temperature and face conditions, the autocorrelation function was fitted into exponential decay in the region of correlation values between 0.01 and 1, and relaxation time was obtained (Fig. 5c). In Layer 1, the primary face showed the lowest relaxation time at the range of 209 –269 K, and the fast transition ratio of the faces corresponds to the large ratio of liquid-like molecules. At a temperature higher than 259 K, the transition is quicker than LDE time, suggesting the mixing behavior of solid and liquid, whereas these states are maintained at low temperatures. Therefore, at high temperatures, the molecular behavior of QLLs can include frequent transitions of the dynamical states, whereas unique QLL molecules may exist at low temperatures such as those at the edges of solid-like clusters.

**Long-ranged orderings of solid- and liquid-like molecules**

The ratio of liquid-like molecules varies widely in the investigated temperature range, and the mobile and immobile molecules form cluster. These clusters are expected to reduce the free energy of the surface, balancing the surface energy of solid-like, liquid-like, and vapor interfaces and entropic effects contributed by surface roughness. Here, we characterize the structure

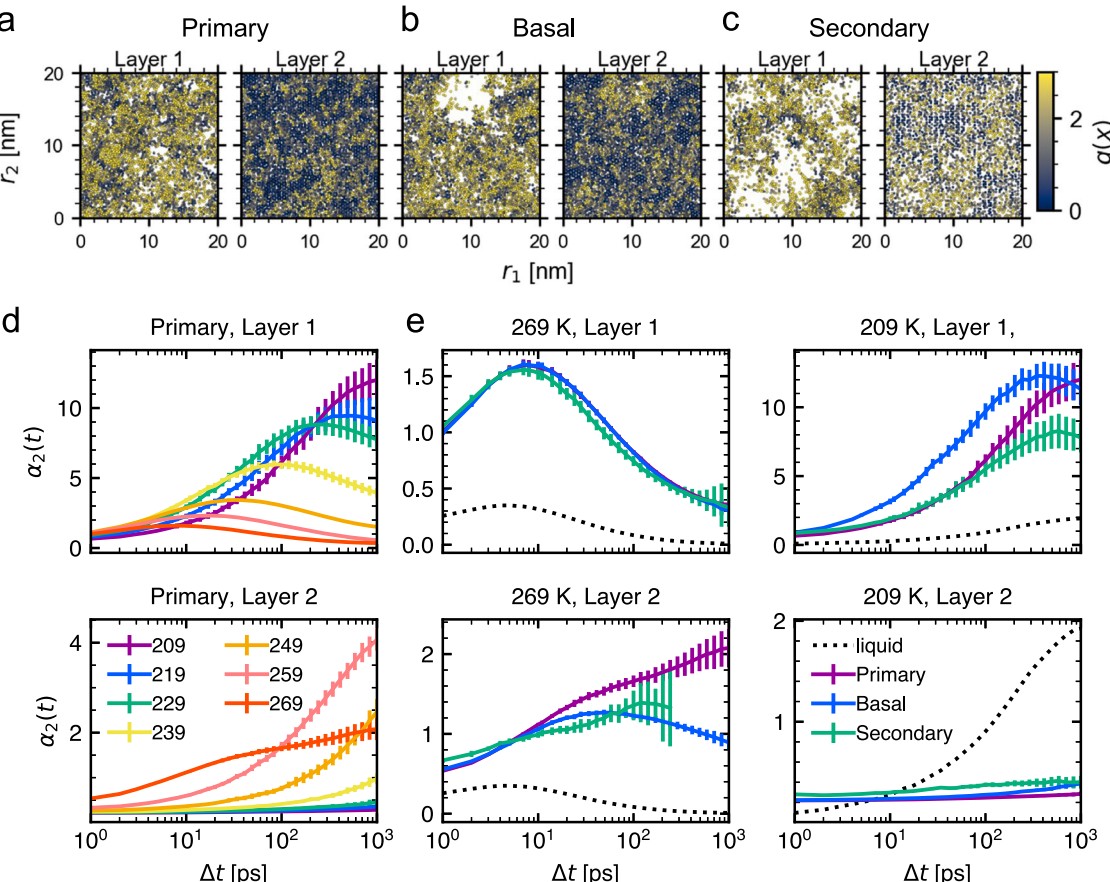

**Fig. 4 | Heterogeneous distributions of mobile molecules. a–c** At 269 K, snapshots of oxygen atoms are colored according to $g(x)$ for (**a**) primary prism, (**b**) basal, and (**c**) secondary prism faces. Immobile atoms are indicated by low $g(x)$ and colored in gray. The axes $(r_1, r_2)$ are $(y, z)$ for primary prism, $(x, z)$ for basal, and $(x, y)$ for secondary prism, respectively. **d** Non-Gaussian parameter $\alpha_2(t)$ of the QLLs for Layer 1 and 2 of the primary prism face at different temperatures in Layers 1 and 2. **e** Non-Gaussian parameter of different faces in Layers 1 and 2 at 209 and 269 K. Error bars indicate the deviation of the mean for time blocks of 5 ns and both interfaces.

of the clusters using the structure factor $S(q)$ and investigate the behavior at low wavevector region for the long-ranged ordering (see Methods for details). As an example, $S(q)$ dictates that liquid-like molecules in Layer 1 of the primary prism have increases of long-range ordering upon cooling until 239 K (Fig. 6a). Yet, since no maximum was observed at this scale, the size of clusters varies without characteristic lengths. This may be due to the limited system size, a characteristic size may appear when simulating larger systems.

To compare the long-range ordering between different temperature and crystal faces, the integrated $S(q)$ was calculated in the range of $0.1 \leq q \leq 0.3$, which corresponds to about 2.09–12.5 nm in the real space[54,55]. The solid-like molecules in Layer 1 exhibited a maximum of around 240 K for all the faces (Fig. 6b). We presume that this is caused by two complementary factors. At low temperatures, heating causes molecular rearrangement, inducing the defects to assemble with increasing cluster sizes of the solid-like structures (Fig. 6c). At high temperatures, the number of solid-like molecules decreases, and thus their cluster size shrinks. Regarding the liquid-like clusters in Layer 1, the bilayer crystals (primary prism and basal faces) showed a similar trend. Likewise, to the solid-like molecules, heating induces molecular rearrangement up to around 240 K. A Decrease of the ordering at temperatures between 240 and 260 K would be owing to the growth of the ratio of liquid-like molecules to largely cover the interfaces. Further heating leads to the formation of small vapor nuclei as indicated by an increase of the structure factor at 269 K. In the secondary prism face, the sparse interface contains solid- and liquid-like molecules and small vapor nucleus at low temperature (Fig. 6d). As indicated by the increase of liquid ordering (Fig. 6b), large vapor nuclei are formed after 259 K.

In Layer 2, the integrated $S(q)$ of solid-like molecules increased upon heating for all investigated temperatures. This would be caused by an increasing number of liquid-like molecules surrounding the solid-like clusters, resulting in long-range orders of the solid-like molecules (Fig. 6e). The liquid-like molecules have a qualitatively identical trend to that in Layer 1 of the bilayer faces. Since the vapor nuclei is not formed, the sudden rise near melting temperature does not happen. These trends are common for the three faces.

Sanchez et al. report sudden changes in vibrational microscopy and molecular orientation at 259 K of bilayer faces (primary prism and basal faces) and propose entire bilayer melting[19]. Concerning this phenomenon, our simulation suggests that the entire melting does not happen at an atomistic scale in Layer 2. Instead, our simulation suggests that clusters of immobile molecules start to crack at 259 K, and mobile molecules move along the boundary of the immobile clusters (Fig. 6e). The immobile clusters are still present at 269 K. Large clusters of solid-like molecules remain more than tens ns scale even at 259 K (see Supplementary Movie 1 for Layer 2 of basal face at 259 K).

## Discussion

The ice surface is covered by a liquid-like film formed in a few molecular layers. Structure-based analyses revealed that at the molecular resolution, the QLL includes an assembly of local liquid- and solid-like structures. However, a dynamic perspective of these clusters remains unclear. Although structural analysis suggests that the QLL is intermediate to the solid and liquid, molecules in it show higher diffusion than those in the bulk[21]. In this study, we determined the dynamical states of single molecules in the QLL

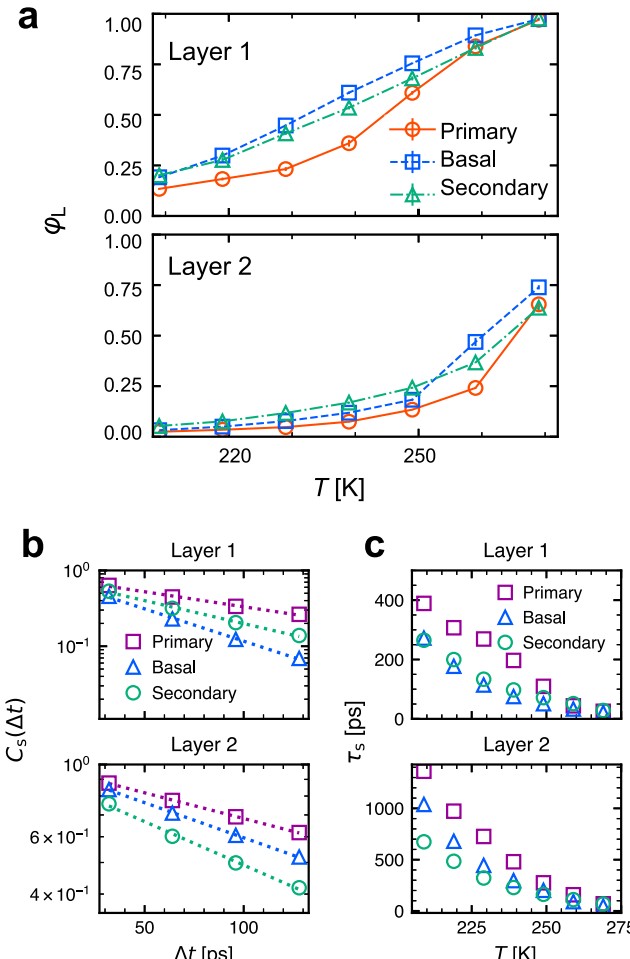

**Fig. 5 | Analysis of liquid and solid-like molecules in QLLs on the three faces at temperature range 209–269 K. a** Liquid ratio $\Phi_L$ at different temperature in Layers 1 and 2 on the three faces. The error bar is shown as the standard deviation of the mean for time sections by 5 ns and both interfaces of the slab were used. **b** Time scale of solid-to-liquid transition in Layer 1 and 2 on the three faces at 249 K. The survival time correlation $C_s$ was calculated for solid-like molecules to stay in the state (see Methods for details). Dashed line is obtained by exponential fitting $C_s(\Delta t) = C_0 \exp(-t/\tau_s)$, where $\tau_s$ is the relaxation time and $C_0$ is a constant. **c** The relaxation time at different temperature.

based on MD simulations and ML analysis. Using simulation data of the bulk water and ice systems, the molecules in QLLs are classified into liquid- and solid-like dynamics. As there are spatially biased distributions of mobile and immobile molecules, the two dynamical state molecules form clusters. We investigated the ratio of liquid-like molecules over and within the QLL layers for different crystal faces in the temperature range 209–269 K. Subsequently, we evaluated the transition rate of molecules between different dynamical states. The results indicated that the dynamical state changed frequently near the melting temperature, and intermediate dynamics could exist at low temperatures. These findings suggest that molecules constituting the solid- and liquid-like domains vary over time, providing insights into the dynamical transitions of QLLs. Finally, we demonstrated that the domains have no characteristic size, but their long-range orderings are sensitive to temperature and the faces.

Surface properties seem to vary among the faces. Regarding the bilayer structures, the underlying geometry appears to have an impact on dynamical heterogeneity in Layer 1 at low temperature or Layer 2 at high temperature. The structure of primary faces includes a clear separation of liquid- and solid-like parts and defects, whereas the basal face shows more disorganized interfaces. It is proposed that the basal is more liquid-like and

homogeneous at low temperatures. Upon heating to 259 K, large parts of the basal face in Layer 2 melt and it mitigates the confinement effect. The tendency of earlier melting in basal faces was also suggested in experiments by Sanchez et al.[19]. One of the characteristics of the secondary prism face is the frequent switching between Layers 1 and 2. Although the in-layer properties of the secondary prism face do not significantly vary from the bilayer structures, the secondary prism surface exhibits differences in some points. For instance, Layer 3 is more fluidic than in the other faces. In Layer 1, the high degree of long ranged-ordering of liquid-like molecules appears near melting point.

Dynamical heterogeneity is known to occur in supercooled liquids in the bulk. Compared to those systems, the premelting water molecules are different in two aspects. First, dynamical heterogeneity of the premelting water system originates from not only the intrinsic properties of water molecules but also the system geometry, whereas the supercooled liquid systems with dynamical heterogeneity can be geometrically homogeneous. In the surface geometry during premelting, the QLLs are confined between the vapor and ice substrate that supports crystallization. As a result of taking the most stable enthalpic and entropic balance of the surface[27], the rough surface with partial melting structure occurs. The crystal-like structure is immobile owing to the large mass of the cluster and its interactions with the substrate layers. Secondly, because water molecules form hydrogen bonds with specific orientations, they tend to produce clear local tetrahedral structures. Previous works detected the local structures of ice using order parameters and machine learning techniques[25,34,56]. In contrast, particles in model glass systems have uniform potential in space, and there is yet no general agreement on the glass structure despite efforts to find the characteristic middle-range structures of glass[57,58]. Therefore, we conclude that the surface geometry and non-uniform potential of ice enhance the dynamical heterogeneity in QLL.

This study has several limitations. First, because the finite system size affects the size of the clusters, much larger systems should be simulated in future research. It is also worth investigating how these atomic-scale inhomogeneities influence surface roughness at the mesoscale and droplet formation at the macroscale. Secondly, the behavior of QLLs depends significantly on the vapor pressure[15,25], which however was not considered here. Especially, since macroscopic structures form in the over- and undersaturation of the vapor[24], sophisticated simulations controlled under over- and under-saturation conditions would provide a better understanding of the QLLs. Finally, the approach we used to distinguish the solid- and liquid-like molecules does not consider the impact of decreased mobility caused by cooling. Because the diffusion coefficient decreases with decreasing temperature, a structure-based classification may be more suitable than a dynamics-based one at lower temperatures. A combined analysis using both structural and dynamical information can be useful for this purpose. Furthermore, because our approach used short-term trajectories (64 ps) to obtain $g(x)$, it was unable to identify highly quick transitions between solid- and liquid-like states.

Heterogeneity of QLLs at the atomic resolution may help researchers understand the properties of ice. In the context of chemistry, guest molecules such as ions and trace gases aggregate on the surface of ice[59]. Hudait et al. indicated that the molecule clusters at liquid-like areas regardless of hydrophilic and hydrophobic molecules, in their work using the mW model[60]. Concerning our work, the longer-ordered liquid region would be crucial and likely to cluster molecules, providing absorption sites for trace gases and enabling their efficient nucleation. If there is only short-range orderings are present, traces of gases would dissipate freely across the surface. In the presence of an adequate scale of the ordering, trace gasses are likely to be confined, allowing them to clutter at a size larger than the nucleus. The all-atom simulation in this scenario will differentiate the nucleation processes of different chemical species. In connection with astrochemistry, ice exists in the installer medium and comets at low temperatures ( < 100 K), and catalytically works for chemical reactions[61]. The surface defects demonstrated at low temperatures (Fig. 6c) possibly function as pores to host the chemical reactions. Moreover, mechanical properties

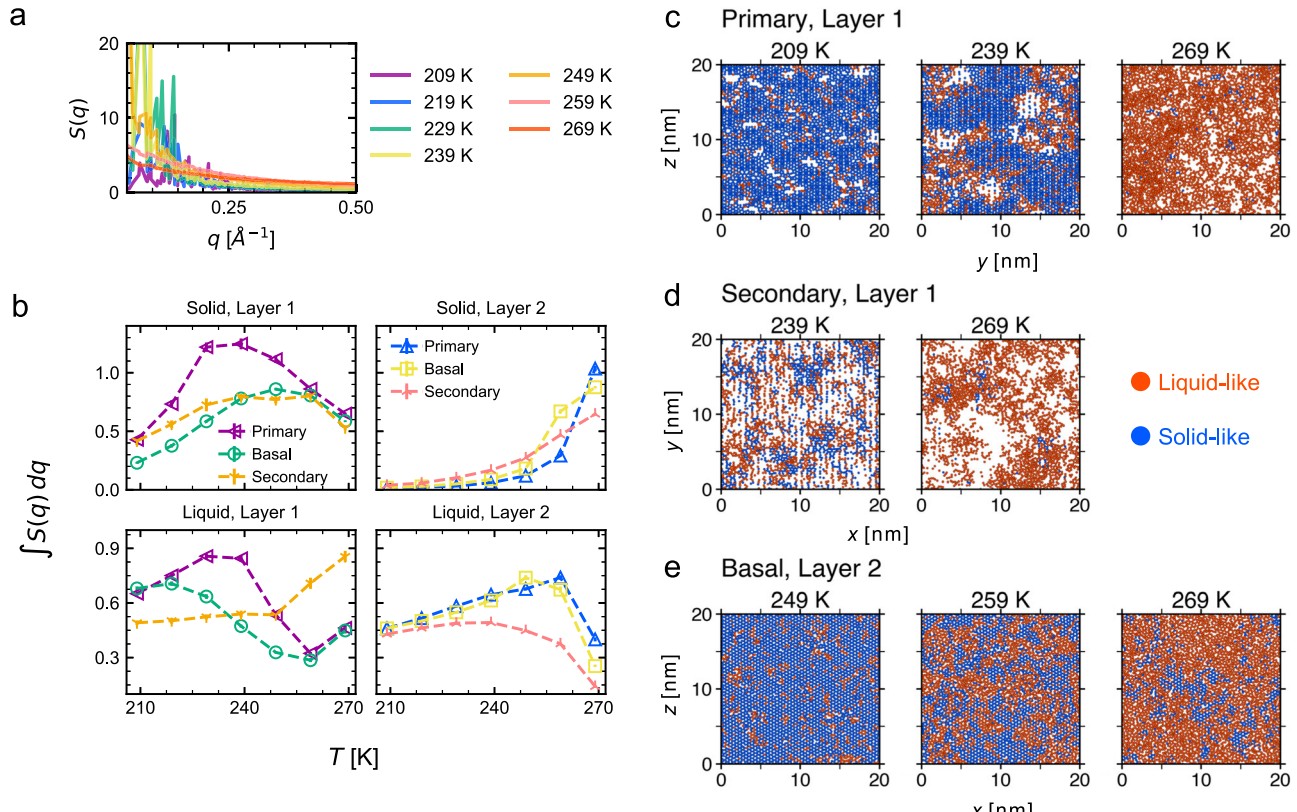

**Fig. 6 | Analysis of long-ranged orderings of solid- and liquid-like molecules.**
**a** Structure factor $S(q)$ calculated on solid-like molecules for Layer 1 of the primary prism face. **b** Integrated $S(q)$ at low $q$ region (0.05 Å$^{-1}$≤$q$≤ 0.3 Å$^{-1}$). Temperature-dependent orderings are compared between the three faces about solid- and liquid-like molecules in Layers 1 and 2. The error bar is shown as the standard deviation of the mean for time sections by 5 ns and both interfaces of the slab was used.
**c–e** Representative snapshots with molecules are colored based on the classified states. Only oxygen atoms are shown.

such as friction originate from molecular diffusion. It is also possible that the surface roughness[62] observed in the second molecular layer is connected to the macroscopic roughness, which makes ice slippery[63].

## Methods
### Molecular dynamics simulations
All-atom MD simulations of large ice/vapor interface systems were performed using the TIP4P/Ice model[43]. This model has shown good agreement in the melting point (269.8 K[64]) with experiments, and it is commonly used to simulate ice and premelting systems. Although it is desired to rigorously treat quantum or many-body interactions for the best prediction of QLL properties as in some models[65,66], the rigid body model in all-atom scale is the proper choice for our study in terms of the computation cost and accuracy. Yet, we note that TIP4P/Ice is justified in the context of ice/water coexistence, and the dynamical properties vary from the experiments[67].

A proton-disordered ice crystal was generated using the open-source software GenIce[68]. The crystal consisted of 432,000 water molecules and had an approximate size of 23.4 nm × 22.0 nm × 27.1 nm. The large crystal size was intended to produce ice/water clusters of approximately 5 nm × 5 nm as observed in a previous study[25]. The $x$, $y$, and $z$ axes are perpendicular to the primary prism, basal, and secondary prism faces, respectively. Because ice crystals deform owing to thermal expansion, we performed anisotropic *NPT* simulation using stochastic velocity rescaling[69] and Berendsen pressure coupling[70] for 1 ns. The initial crystal structure was rescaled using the average unit cell length obtained from the *NPT* simulation. The slab system was created from the rescaled crystal by extending one direction of the unit cell by 100 nm. The simulations were performed in the *NVT* ensemble for 150 ns with initial velocities given randomly. Trajectories of the two interfaces in the last 50 ns were analyzed. A cutoff distance of 0.85 nm was applied for both

the van der Waals and short-range electrostatic interactions. Long-range electrostatic interactions were calculated by the smooth particle mesh Ewald method[71], using fourth-order interpolation with a spacing of 0.12 nm. The water structure was fixed using the SETTLE algorithm[72]. Periodic boundary conditions were applied in all the directions. The equations of motion were integrated by 2 fs, and the trajectories were output every 1 ps. All simulations were conducted using GROMACS 2020.6[73].

We also prepared bulk hexagonal ice and supercooled water to compare the molecular dynamics between the bulk systems and QLLs. The ice crystal contained 8192 molecules, and the supercooled water system was created by melting the ice crystal at 400 K in *NVT* ensemble. The corresponding MD calculations were performed under the same conditions as those used for the QLL system. Regarding the combing dataset used in ML, individual solid/liquid simulations are combined, i.e. 16,384 molecules are contained in this dataset.

### Liquid/Solid classification of dynamical states using machine learning
By performing dimension reduction to MD trajectories, a single molecular trajectory was represented as a scalar value $g(\mathbf{x})$. $g(\mathbf{x})$ was previously used to analyze multiple systems and determine the characteristic MD trajectories[48,74,75] (please refer to Supplementary Methods and these works for descriptions of $g(\mathbf{x})$). Assuming that the distribution of $g(\mathbf{x})$ based on short-term trajectories shows significant difference between solid and liquid states, a threshold $\alpha$ for discriminating these two states can be defined as,

$$X(\mathbf{x}) = \begin{cases} \text{Solid}, & \text{if } g(\mathbf{x}) \leq \alpha \\ \text{Liquid} & \text{otherwise} \end{cases} \quad (2)$$

where $X$ denotes the state of the molecule. Likewise, $g(\mathbf{x})$ was obtained for the premelting trajectory $\mathbf{x}_p$,

$$g(\mathbf{x}_p) = \mathbb{E}_{\mathbf{x}' \sim \mathbf{y}'}[f^*(\mathbf{x}_p) - f^*(\mathbf{x}')] \tag{3}$$

where $\mathbf{x}'$ is the trajectory sampled from the solid system with probability distribution $\mathbf{y}'$. The solid/liquid classification was performed.

$$X(\mathbf{x}_p) = \begin{cases} \text{Solid}, & \text{if } g(\mathbf{x}_p) \leq \alpha \\ \text{Liquid}, & \text{otherwise} \end{cases} \tag{4}$$

The layer of short-term trajectories is determined based on the mean position during the 64 ps. Because of the low uncertainty of classification due to the model parameter, we used one of the trained models to calculate $g(\mathbf{x})$.

## Survival time correlation

For a molecule $j$ in layer $\alpha$, the binary function $p_{\alpha,j}(t, t + t; t_0)$ is defined so that it takes 1 when the molecule's residence from time $t$ to $t + \Delta t$ without leaving the layer longer than time $t_0$. For a lag time $\Delta t$, the survival time correlation $C(\Delta t)$ is defined as,

$$C(\Delta t) = \frac{1}{N} \sum_{j=1}^{N} \frac{1}{t_{run} - \Delta t} \sum_{t=0}^{t_{run} - \Delta t} p_{\alpha,j}(t, t + \Delta t; t_0) \tag{5}$$

where $N$ is the number of water molecules, $t_{run}$ is the simulation time step, and $t_0$ is the simulation ouput interval. In an ideal situation where the layer transition occurs at a constant ratio to the number of surviving molecules, $C(t)$ exhibits exponential decay and the inverse of decay time constant is the relaxation time.

Similarly, the survival time correlation to remain in a dynamical state was estimated by modifying the binary function above. The new binary function takes a value of 1 when the molecules $j$ stays in the state from time $t'$ to $t' + \Delta t$ without leaving the initial state longer than time $t_0$. Molecules jumping into the other layers are removed from the analysis.

## Non-Gaussian parameter

The dynamical heterogeneity can be described using the non-Gaussian parameter[50]. The mean square displacement of $N$ particles is defined as,

$$\langle r^2(\Delta t) \rangle = \left\langle \frac{1}{N} \sum_{i=1}^{N} ||\mathbf{r}(t + \Delta t) - \mathbf{r}(t)||^2 \right\rangle, \tag{6}$$

and the mean of four-power displacement as,

$$\langle r^4(\Delta t) \rangle = \left\langle \frac{1}{N} \sum_{i=1}^{N} ||\mathbf{r}(t + \Delta t) - \mathbf{r}(t)||^4 \right\rangle. \tag{7}$$

If particles show Brownian motion, the distribution of square displacements is Gaussian. The non-Gaussian parameter describes the degree of the deviation from the Gaussian and is defined as

$$\alpha_2(\Delta t) = \frac{3\langle \Delta r^4(\Delta t) \rangle}{5\langle \Delta r^2(\Delta t) \rangle^2} - 1 \tag{8}$$

In the calculation of the mean square displacement, positions of oxygen $\mathbf{r}$ are considered only in the in-plane coordinate. Molecules leaving the layer within $\Delta t$ were removed from the analysis. The bracket $\langle \cdot \rangle$ is average over observation time.

## Structure factor

The long-range ordering was evaluated using the structure factor[76]. The 2D structure factor is useful in the case of interfaces,

$$S(\mathbf{q}) = \left\langle \frac{1}{N} \left[ \sum_{i=1}^{N} \sin(\mathbf{q} \cdot \mathbf{r}) \right]^2 \right\rangle + \left\langle \frac{1}{N} \left[ \sum_{i=1}^{N} \cos(\mathbf{q} \cdot \mathbf{r}) \right]^2 \right\rangle \tag{9}$$

where $N$ is the number of particles, $\mathbf{r}$ is the oxygen position in the layers, and the wave vector $\mathbf{q}$ is denoted using integers $n_1$, $n_2$ and unit cell length $L_1$ and $L_2$ as $\mathbf{q} = (2\pi n_1/L_1, 2\pi n_2/L_2)$. In an isotropic system, $S(\mathbf{q})$ is a function of the length of wavelength $q$.

## Data availability

Initial and final configurations of MD simulations, the simulation input files, and numerical data for Figs. 1d, e, 2b–d, 3a–c, 4d, e, 5a–c, and 6a, b are stored in Figshare data repository at https://doi.org/10.6084/m9.figshare.25649127[77]. Simulation movie of the basal face of Layer 2 at 259 K is available in the file Supplementary Movie 1.

## Code availability

Codes of MD analysis are available from the corresponding author upon reasonable request. Pseudo-code code to calculate g(x) is available in Supplementary Software 1, which is included in Supplementary Information. Machine learning codes to calculate g(x) are covered by a patent (Patent applicant: Keio University. Inventors: K. Yasuoka, D. Yuhara, K. Endo, and K. Tomobe. Application number: JP.2019048988.A. Status of application: published unexamined patent application), and so are not publicly available.

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

## Acknowledgements

This work was supported by JST, CREST Grant No. JPMJCR2093, Japan. The authors are grateful to Prof. K. Kurihara (Tohoku University), Prof. M. Mizukami (Tohoku University) for helpful discussions and useful comments. This work was supported in part by MEXT as "Program for Promoting Research on the Supercomputer Fugaku" grant number JPMXP1020230325, and the computations were partially carried out using the supercomputer Fugaku provided by the RIKEN Center for Computational Science. I.Y. is supported by Grant-in-Aid for JSPS Fellows Grant No. JP23KJ1918.

## Author contributions

K.Y., N.A. conceptualized the research. I.Y. and N.A. performed the simulations. I.Y analyzed the data. I.Y. wrote manuscript and N.A., K.E. and K.Y. edited the manuscript.

## Competing interests

The authors declare no competing interests.
