## [Peer Review File · Communications Chemistry]

Reviewers' comments:

Reviewer #1 (Remarks to the Author):

The manuscript by Yasuoka and coworkers reports a classical molecular dynamics (MD) study of the surface of ice from 209 to 269 K, employing the TIP4P/Ice empirical force field. The authors simulate very large models of the three low index facets of hexagonal ice to investigate heterogeneity in the dynamics of the quasi-liquid layer (QLL). For the analysis they generated a Machine Learned descriptor that identifies solid and liquid molecules using a dimensional reduction to measure the dynamical differences (specifically for diffusion of oxygen and hydrogen and dipole rotation). At low temperature, this descriptor allows one to identify a dynamically unique state for the molecules at the QLL, while at high temperature dynamical heterogeneity occurs at the QLL with liquid and solid domains, and frequently switching between states. They find no characteristic domain size but find long-range ordering depends on temperature and crystal face.

This work is potentially interesting: the analysis tools and the results are original. The size of the models is suitable for this type of study, but the simulation time is definitely insufficient. It is very unlikely that 10 ns of MD are sufficient to equilibrate the surface of ice: for comparison, in Ref. 19 the slabs were equilibrated for at least 50 ns, and in Ref. 26 MD simulations were beyond 300 ns. Similarly, 5 ns of production runs are too short, as relevant dynamical processes at the surface of ice have been observed over the 10 ns time-scale.

This issue may affect the validity of the main results of this paper, and a careful investigation of time convergence is imperative.

There are few other minor issues:

-Is Figure 6c) missing the Integrated $S(q)$ for liquid like molecules in layer 2? Also unclear in Figure 6b) if the coloring is based on $S(q)$ or $g(x)$ as it was in Figure 4a. Maybe should add a colorbar.

-What temperatures are the snapshots from in Figure 4a)b)c)

-Page 3 ~ 15 lines down. "For the oxygen atom over 64 ps (Fig. 4b)" should this be Fig 2b?

-Figure 3a)b)c) should include in figure caption what temperature is being shown.

-There is no mention of how large the vacuum layer is. Additionally, I assume the authors exposed two surfaces to vacuum, if this is the case, did they sample from both interfaces to obtain more statistics?

-Page 5 column 2 about 20 lines down they say, "At high temperatures, the number of liquid-like molecules decreases, and so their cluster size shrinks" This does not sound right if referred to Figure 4.

-Ref 19 discusses that the QLL exhibits a layer-by-layer pre-melting. It would be interesting to make a connection to the differences in inhomogeneous dynamics at different temperatures for Layers 1 and 2 to the premelting transition at -10 K where the onset of the pre-melting of the 2nd bilayer is found to occur. For example, in Figure 4d) it can be clearly seen that there is a transition at 259 K in the α_2 distribution which happens at the same temperature at which pre-melting starts for the 2nd bilayer.

Reviewer #2 (Remarks to the Author):

In this manuscript, the authors investigate the properties of the quasi-liquid layer (QLL) of ice using extensive molecular dynamics (MD) simulations carried out with the TIP4P/ice model, which are subsequently analyzed using a machine learning (ML) algorithm to identify possible dynamical heterogeneities as a function of temperature and depth from the surface. Understanding the properties of the QLL at the molecular level has wide-ranging implications in environmental science, meteorology, and material engineering. This study provides a different perspective of the properties of the QLL, which can be relevant to these fields.

My concern is about the robustness of the analyses and results reported in this study. While the use of the TIP4P/Ice model is justified in the context of simulating ice/water coexistence, concerns arise regarding its limitations in representing supercooled water and certain dynamical properties of water, as noted in recent publications [J. Chem. Phys. 158, 204505 (2023); J. Chem. Phys. 158, 064503 (2023)]. The potential impact of these limitations on the results and conclusions presented in this manuscript warrants attention. This prompts the question of whether the MB-pol model, which shows remarkable accuracy in predicting the properties of water across all phases [e.g., J. Chem. Phys. 145, 194504 (2016); J. Phys. Chem. Lett. 13, 3652 (2022); Nat. Commun. 14, 3349 (2023)], could have been a more robust choice. The use of MB-pol might have provided additional credibility to the findings and reinforced the robustness of the analyses. The authors are encouraged to consider these aspects to further enhance the reliability of their simulations.

Given the study's strong alignment with the fields of physical chemistry and chemical physics, it might find a more focused and appreciative audience in journals that are traditionally centered on these topics, such as the Journal of Chemical Physics or the Journal of Physical Chemistry C. While the choice of publication is ultimately at the discretion of the authors and the Editor, such journals could potentially offer greater visibility and impact within the relevant scientific community, ensuring the work reaches an audience well-versed in the properties of ice and molecular simulations.

In conclusion, this manuscript provides valuable insights into the dynamic heterogeneity of quasi-liquid layers on ice surfaces. It opens up new avenues for research in the field and presents a nice effort in bridging gaps in our understanding of ice physics. Considering the point raised about the choice of the water model, this work promises to contribute new insights into the properties and behavior of the QLL of ice.

Reviewer #3 (Remarks to the Author):

The paper by Yausoka et al. is an interesting if quite fundamental study of water ice quasi liquid layers. The others report fluctuating heterogeneity on three different faces of a particular type of ice crystal (of which there are many).

I am not totally convinced that this paper necessarily merits publication in *commschem*. It is novel, well executed and fairly well written, but most of the novelty (written in its current form) comes from the application of ML and less so on the qualitative findings about the nature of the surfaces themselves. There are some other issues as well:

1) The simulations for each surface appear to be 5ns unless I have misunderstood? What justification do the authors have that all the rare events of interest, the dynamic heterogeneity, and all properties of interest are converged/sampled frequently enough over 5ns. It feels reasonably short for such a complex interplay of physical states.

2) Can the authors explain what the need for machine learning is in this study? The ML work seems well executed and it's good to see new and efficient techniques being applied but what is the obstacle to brute force analysis? I may have missed it but this aspect of the work does not seem to have been explained/justified?

3) Related to 2 - what qualitative differences in the interpretation of the results/analysis if the input to ML is e.g. 32ps (see figure 2c) and 128ps? How confident are the authors that a ~10% error does not lead to chemically meaningful differences in interpretation?

4) How does the dynamic heterogeneity affect chemically relevant properties? Surface roughness, availability and number and strength of adsorption sites for trace gases (atmospheric catalysis and kinetics), lifetime of solid like regions. How does it vary for each surface? I appreciate some of these questions do not have precise answers and some of the answers can be inferred from the presented data but I would prefer to see, as a chemist, a more reaction chemistry focused or aware interpretation of the findings.

Reviewers' comments:

Reviewer #1 (Remarks to the Author):

The manuscript by Yasuoka and coworkers reports a classical molecular dynamics (MD) study of the surface of ice from 209 to 269 K, employing the TIP4P/Ice empirical force field. The authors simulate very large models of the three low index facets of hexagonal ice to investigate heterogeneity in the dynamics of the quasi-liquid layer (QLL). For the analysis they generated a Machine Learned descriptor that identifies solid and liquid molecules using a dimensional reduction to measure the dynamical differences (specifically for diffusion of oxygen and hydrogen and dipole rotation). At low temperature, this descriptor allows one to identify a dynamically unique state for the molecules at the QLL, while at high temperature dynamical heterogeneity occurs at the QLL with liquid and solid domains, and frequently switching between states. They find no characteristic domain size but find long-range ordering depends on temperature and crystal face.

This work is potentially interesting: the analysis tools and the results are original. The size of the models is suitable for this type of study, but the simulation time is definitely insufficient. It is very unlikely that 10 ns of MD are sufficient to equilibrate the surface of ice: for comparison, in Ref. 19 the slabs were equilibrated for at least 50 ns, and in Ref. 26 MD simulations were beyond 300 ns. Similarly, 5 ns of production runs are too short, as relevant dynamical processes at the surface of ice have been observed over the 10 ns time-scale.

This issue may affect the validity of the main results of this paper, and a careful investigation of time convergence is imperative.

We thank the reviewer for their interest and for pointing out the issue of simulation time. Following their comment, we have extended our simulations to a total of 150 ns where the first 100 ns is for equilibration and the last 50 ns for sampling. We also revised a minor point of our simulation, i.e. cell size was determined by the average cell length in the pre-performed *NPT* simulation. The new data analysis revealed that the key findings are largely unchanged from the previous version. The convergence of simulation was checked based on the standard deviation of the mean values of time blocks, as shown by error bars in Figs 1e 3a–c, 4d and e, 5a and 6b. The time blocks are 5 ns time sections of the trajectories from both interfaces, thus 20 samples are used to estimate the deviation.

In the revised manuscript, the simulation length is described as

“The simulations were performed in the *NVT* ensemble for 150 ns with initial velocities given randomly. Trajectories of the two interfaces in the last 50 ns were analyzed.” (page 16)

There are few other minor issues:

-Is Figure 6c) missing the Integrated $S(q)$ for liquid like molecules in layer 2? Also unclear in Figure 6b) if the coloring is based on $S(q)$ or $g(x)$ as it was in Figure 4a. Maybe should add a colorbar.

We thank the reviewer for pointing this out. We have added $S(q)$ for liquid-like molecules in layer 2 (Fig. 6b). Also, in the updated version, molecules are colored in a binary way to clarify the solid and liquid-like patterning (Fig. 6c–e).

FIG. 6. Analysis of long-ranged orderings of solid- and liquid-like molecules. (a) Structure factor $S(q)$ calculated on solid-like molecules for Layer 1 of the primary prism face. (b) Integrated $S(q)$ at low q region ($0.05 \text{ \AA}^{-1} \leq q \leq 0.3 \text{ \AA}^{-1}$). Temperature-dependent orderings are compared between the three faces about solid- and liquid-like molecules in Layers 1 and 2. (c–e) Representative snapshots with molecules are colored based on the classified states. Only oxygen atoms are shown.

-What temperatures are the snapshots from in Figure 4a)b)c)

These snapshots are obtained at just below the melting point 269 K. We have clarified this information in the figure caption.

We have updated the manuscript as:

“(a–c) At 269 K, snapshots of oxygen atoms are colored according to $g(x)$ for (a) primary prism, (b) basal, and (c) secondary prism faces.” (page 29)

-Page 3 ~ 15 lines down. “For the oxygen atom over 64 ps (Fig. 4b)” should this be Fig 2b?

We thank the referee for this pointing out, and we have corrected this point.

We have updated the manuscript as:

“their short-term trajectories were evaluated using $g(x)$ for the translational dynamics of oxygen atom over 64 ps (Fig. 2b)” (page 6)

-Figure 3a)b)c) should include in figure caption what temperature is being shown.

These snapshots are obtained at just below the melting point 269 K. We have clarified this information in the figure caption.

In the caption, a sentence is added

“FIG. 3. Analysis of dynamics of oxygen molecules in QLLs at 269 K.” (page 28)

-There is no mention of how large the vacuum layer is. Additionally, I assume the authors exposed two surfaces to vacuum, if this is the case, did they sample from both interfaces to obtain more statistics?

We apologize for the lack of simulation descriptions. The thickness of the vacuum layer is 100 nm, which is long enough to prevent the molecules from getting out of one side into the vapor phase and immediately reaching the other exposed face.

Based on the comment by the reviewer, we used both interfaces in this revision to increase the statistics. To clarify the statistics, we divided the trajectories into time blocks of 5 ns. Over the time blocks in both interfaces, we calculated the deviation of the mean of each time block.

“The slab system was created from the rescaled crystal by extending on the direction of the unit cell by 100 nm.” (page 15)

-Page 5 column 2 about 20 lines down they say, “At high temperatures, the number of liquid-like molecules decreases, and so their cluster size shrinks” This does not sound right if referred to Figure 4.

We apologize for our typo, and corrected “liquid-like molecules” to “solid-like molecules”.

“At high temperatures, the number of solid-like molecules decreases, and so their cluster size shrinks” (page 12)

-Ref 19 discusses that the QLL exhibits a layer-by-layer pre-melting. It would be interesting to make a connection to the differences in inhomogeneous dynamics at different temperatures for Layers 1 and 2 to the premelting transition at -10 K where the onset of the pre-melting of the 2nd bilayer is found to occur. For example, in Figure 4d) it can be clearly seen that there is a transition at 259 K in the α_2 distribution which happens at the same temperature at which pre-melting starts for the 2nd bilayer.

We thank the reviewer for pointing this out. Sanchez et al. report sudden changes in vibrational microscopy and molecular orientation at 259 K of bilayer faces (primary prism and basa faces) and propose entire bilayer melting [*PNAS*, 114, 227–232 (2017)]. Concerning this phenomenon, our simulation suggests that the entire melting does not happen at an atomistic scale in Layer 2 (please refer to Fig. 6c attached above). Instead, our simulation suggests that clusters of immobile molecules start to crack at around 259 K and the confinement effect by solid-like regions is alleviated. The precise temperature seems lower in the basal than primary face. The immobile clusters are still present at 269 K as well. Non-continuous changes in Layer 2 are also supported by the layer transition kinetics (Fig. 1e), a decrease in dynamical heterogeneity (Fig. 4d), and liquid-like ratio (Fig.5a).

In the light of the pointing out, we added sentences in the revised manuscript:

“This trend of the dynamical heterogeneity can be interpreted as the effect of confinement is significant until a certain point between 259 and 269 K, and further heating alleviates the confinement effect” (page 9)

“Sanchez et al. report sudden changes in vibrational microscopy and molecular orientation at 259 K of bilayer faces (primary prism and basal faces) and propose entire bilayer melting [18]. Concerning this phenomenon, our simulation suggests that the entire melting does not happen at an atomistic scale in Layer 2. Instead, our simulation suggests that clusters of immobile molecules start to crack at 259 K, and mobile molecules move along the boundary of the immobile clusters (Fig. 6d). The immobile clusters are still present at 269 K. Large clusters of solid-like molecules remain more than tens ns scale even at 259 K (Supplementary Movie 1).” (page 12)

Reviewer #2 (Remarks to the Author):

In this manuscript, the authors investigate the properties of the quasi-liquid layer (QLL) of ice using extensive molecular dynamics (MD) simulations carried out with the TIP4P/ice model, which are subsequently analyzed using a machine learning (ML) algorithm to identify possible dynamical heterogeneities as a function of temperature and depth from the surface. Understanding the properties of the QLL at the molecular level has wide-ranging implications in environmental science, meteorology, and material engineering. This study provides a different perspective of the properties of the QLL, which can be relevant to these fields.

My concern is about the robustness of the analyses and results reported in this study. While the use of the TIP4P/ice model is justified in the context of simulating ice/water coexistence, concerns arise regarding its limitations in representing supercooled water and certain dynamical properties of water, as noted in recent publications [J. Chem. Phys. 158, 204505 (2023); J. Chem. Phys. 158, 064503 (2023)]. The potential impact of these limitations on the results and conclusions presented in this manuscript warrants attention. This prompts the question of whether the MB-pol model, which shows remarkable accuracy in predicting the properties of water across all phases [e.g., J. Chem. Phys. 145, 194504 (2016); J. Phys. Chem. Lett. 13, 3652 (2022); Nat. Commun. 14, 3349 (2023)], could have been a more robust choice. The use of MB-pol might have provided additional credibility to the findings and reinforced the robustness of the analyses. The authors are encouraged to consider these aspects to further enhance the reliability of their simulations.

We understand the cornering about the model from the reviewer.

TIP4P/ice model was validated for ice/water coexistence, showing the best agreement among the rigid water models. Yet, the dynamical properties are underestimated as indicated by the proposed paper [JCP, 158, 204505 (2023)]. The paper also suggests that the properties of different water models are consistent in the reduced scale. Since our models use relative differences between the solid and liquid phases of ice, the model effect might be alleviated. Nevertheless, it is worth mentioning that one should note the underestimation by the rigid water model when comparing the absolute value of transport properties to experiments.

To treat the QLL layers rigorously, quantum-level models may be helpful, one of which is the MB-pol model [Nat. Commun., 14, 3349 (2023)], as suggested by the reviewer. MB-pol model has been justified in the context of various ice morphologies, and the detailed treatment of many-body interactions may help the prediction of dynamical properties. We are looking forward to seeing the validation in terms of the transport properties in future work.

Regarding this point, we added a sentence in the revised manuscript.

“Although it is desired to rigorously treat quantum or many-body interactions for the best prediction of QLL properties as in some models [58, 59], the rigid body model in all-atom scale is the proper choice for our study in terms of the computation cost and accuracy. Yet, we note that TIP4P/ice is justified in the context of ice/water coexistence, and the dynamical properties vary from the experiments [60]” (page 15)

Given the study's strong alignment with the fields of physical chemistry and chemical physics, it might find a more focused and appreciative audience in journals that are traditionally centered on these topics, such as the Journal of Chemical Physics or the Journal of Physical Chemistry C. While the choice of publication is ultimately at the discretion of the authors and the Editor, such journals could potentially offer greater visibility and impact within the relevant scientific community, ensuring the work reaches an audience well-versed in the properties of ice and molecular simulations.

In conclusion, this manuscript provides valuable insights into the dynamic heterogeneity of quasi-liquid layers on ice surfaces. It opens up new avenues for research in the field and presents a nice effort in bridging gaps in our understanding of ice physics. Considering the point raised about the choice of the water model, this work promises to contribute new insights into the properties and behavior of the QLL of ice.

We thank the reviewer for their positive comments and suggestions.

Reviewer #3 (Remarks to the Author):

The paper by Yausoka et al. is an interesting if quite fundamental study of water ice quasi liquid layers. The others report fluctuating heterogeneity on three different faces of a particular type of ice crystal (of which there are many).

We appreciate the review's interest.

I am not totally convinced that this paper necessarily merits publication in *commschem*. It is novel, well executed and fairly well written, but most of the novelty (written in its current form) comes from the application of ML and less so on the qualitative findings about the nature of the surfaces themselves. There are some other issues as well:

1) The simulations for each surface appear to be 5ns unless I have misunderstood? What justification do the authors have that all the rare events of interest, the dynamic heterogeneity, and all properties of interest are converged/sampled frequently enough over 5ns. It feels reasonably short for such a complex interplay of physical states.

Thank you for pointing this out. In the previous results, our simulations are continued only for 5 ns after 10 ns of equilibration, which would be too short to sample all the rare events as pointed out. We have extended our simulation to 150 ns, where the last 50 ns was used for analysis. We report largely unchanged results on our conclusion on the ice-vapor interface properties.

In the revised manuscript, the simulation length is described as

“The simulations were performed in the NVT ensemble for 150 ns with initial velocities given randomly. Trajectories of the two interfaces in the last 50 ns were analyzed.” (page 15)

2) Can the authors explain what the need for machine learning is in this study? The ML work seems well executed and it's good to see new and efficient techniques being applied but what is the obstacle to brute force analysis? I may have missed it but this aspect of the work does not seem to have been explained/justified?

We thank the reviewer for pointing this out. In the analysis of water molecules on interfaces in simulations using TIP4P/Ice models, local structures are commonly classified to be solid or liquid based on the order parameters [eg. *Sci. Adv.*, 6, eaay9322 (2020), *Commun. Mater.*, 4, 33(2023)]. However, as the connection between structure and dynamics is not taken for granted, we would prefer to employ the analytical method to investigate the heterogeneity of dynamics. One of the brute force analyses in dynamics would be based on mean square displacements (MSD). There are two notable points to be addressed (1) It is difficult to distinguish the dynamics of supercooled water at least based on the MSD, and (2) high-time resolution classification is required due to frequent layer transition. Our approach uses whole trajectories in time windows. This results in a more accurate classification than the brute-force MSD approach, as tested for the bulk of solid and liquid molecules at 269 K. We believe our ML approach can provide a highly time-resolved description of dynamical states.

To clarify the need for our ML approach, we added sentences

“In the analysis of water molecules on interfaces in simulations using TIP4P/Ice models, local structures are commonly classified to be solid or liquid based on the order parameters [26, 43]. Recently, machine-learning approaches to detect local ice-like structures have been developed [29–31, 44]. However, as the connection between structure and dynamics is not taken for granted, we would prefer to employ the analytical method to investigate the heterogeneity of dynamics. The dynamic properties of a single water molecule include its rotational [45] and translational motions [34, 35]. One of the brute force analyses in dynamics would be based on mean square displacements (MSD). There are two notable points to be addressed (1) It is difficult to distinguish the dynamics of supercooled water at least based on the MSD, and (2) high-time resolution classification is required due to frequent layer

transition. Here, we developed an ML scheme to classify the state of a single molecule as either solid or liquid based on its short-term dynamics (Fig. 2a)” (page 5)

3) Related to 2 - what qualitative differences in the interpretation of the results/analysis if the input to ML is e.g. 32ps (see figure 2c) and 128ps? How confident are the authors that a ~10% error does not lead to chemically meaningful differences in interpretation?

We appreciate the reviewer’s comment on the meaning of the accuracy of the ML method. When using a shorter period 32 ps, there is supposed to be more confusion between solid and liquid molecules. This would lead to the mixing of the two states and would not be beneficial to clarify the transition of the two states (Fig. 5b and c), being affected by the transition rate induced by noise. Also, the long-range ordering may be reduced because the mixing of the two states would occur (Fig. 6b). When using a shorter period of 128 ps, this would lead to confusion of layers, i.e. the properties of Layer 1 and 2 would be similar. Although analysis using a longer period could be practical by removing the molecules that transit layers, it results in the removal of large parts of molecules.

We are confident that our ML method can distinguish the differences in dynamics with high accuracy. This is based on the error rate between independent ML classifications is small. Instead of thinking that the 10% error comes from the ML, it would mean the deviation of dynamics to the structure. In terms of the structure, the local molecular structures up to the five closest molecules almost perfectly predict the phase [*JCP*, 159, 064103 (2023)]. Liquid-like regions seem to be a key factor for absorption and catalysis in the interface of ice, a simulation study by Hudait et al. reports the density of ions absorbed in the surface sublinearly increases the density of the liquid-like molecules [*J. Am. Chem. Soc.*, 139, 10095–10103 (2019)]. Although there should be a strong correlation between the dynamics and solid, we are uncertain about whether dynamics and structure have a direct causal relationship to the surface properties of ice. Further research is needed in this direction.

In the revised manuscript, we added sentences;

“Since the error bar from individual ML runs is minor, the classification error originates from the intrinsic nature of the MD system.” (page 6)

“When using a shorter period 32 ps, there is supposed to be more confusion between solid and liquid molecules. This would lead to the mixing of the two states and would not be beneficial to clarify the transition of the two states, being affected by the transition rate induced by noise. When using a shorter period of 128 ps, this would lead to confusion of layers, i.e. the properties of Layer 1 and 2 would be similar. Although analysis using a longer period could be practical by removing the molecules that transit layers, it results in the removal of large parts of molecules.” (page 6–7)

4) How does the dynamic heterogeneity affect chemically relevant properties? Surface roughness, availability and number and strength of adsorption sites for trace gases (atmospheric catalysis and kinetics), lifetime of solid like regions. How does it vary for each surface? I appreciate some of these questions do not have precise answers and some of the answers can be inferred from the presented data but I would prefer to see, as a chemist, a more reaction chemistry focused or aware interpretation of the findings.

We thank the reviewer for posing the interesting points. We have strong interests in the proposed topics. For other molecules of ions and trace gases, the prominent effect of surface heterogeneity is their aggregation [*Atmos. Chem. Phys.*, 3, 1587–1633 (2014)]. Hudait et al. indicated that the molecule clusters at liquid-like areas regardless of hydrophilic and hydrophobic molecules, in their work using the mW model [*J. Am. Chem. Soc.*, 139, 10095–10103 (2019)]. Concerning our work, the longer-ordered liquid region would be crucial and likely to cluster molecules, providing absorption sites for trace gases and enabling their efficient nucleation. If there is only short-range orderings are present, traces of gases would

dissipate freely across the surface. In the presence of an adequate scale of the ordering, trace gasses are likely to be confined, allowing them to clutter at a size larger than the nucleus. In our future work, we would work on the all-atom simulation in this scenario to differentiate the nucleation of different chemical species, which would contribute to explaining the ratio of trace gases in the air.

Regarding the lifetime of solid-like regions, we found that the larger clusters remain more than tens ns scale even at 259 K (see Supplementary Movie). They undergo translation and replacement of the belonging molecules on the edge. While we assume the solid-like regions are stable, the accurate prediction for cluster lifetime requires more sampling, and to be studied in further research.

Surface properties seem to vary among the faces. Regarding the bilayer structures, the underlying geometry appears to have an impact on dynamical heterogeneity in layer 1 at low temperature or layer 2 at high temperature. The structure of primary faces includes a clear separation of crystalline and defects, whereas the basal face's interface is more disorganized. It is proposed that the basal is more liquid-like and homogeneous at low temperatures. Upon heating to 259 K, large parts of the basal face in Layer 2 melt and it mitigates the confinement effect. The tendency of earlier melting in basal faces was also suggested in experiments by Sanchez et al [*PNAS*, 114, 227–232 (2016)]. One of the characteristics of the secondary face is the frequent switching between Layers 1 and 2. Although the interlayer properties of the secondary prism face do not significantly vary from the bilayer faces, the secondary prism surface exhibits differences in some points. For instance, Layer 3 is more fluidic than in the other faces. In Layer 1, the long ranged-ordering of liquid-like molecules would be related to the surface roughness.

Based on the comment we added sentences,

“In the context of astrochemistry, guest molecules such as ions and trace gases aggregate on the surface of ice [56]. Hudait et al. indicated that the molecule clusters at liquid-like areas regardless of hydrophilic and hydrophobic molecules, in their work using the mW model [57]. Concerning our work, the longer-ordered liquid region would be crucial and likely to cluster molecules, providing absorption sites for trace gases and enabling their efficient nucleation. If there is only short-range orderings are present, traces of gases would dissipate freely across the surface. In the presence of an adequate scale of the ordering, trace gasses are likely to be confined, allowing them to clutter at a size larger than the nucleus. The all-atom simulation in this scenario will differentiate the nucleation of different chemical species.” (page 14-15)

“Large clusters of solid-like molecules remain more than tens ns scale even at 259 K (see Supplementary Movie).” (page 12)

“Surface properties seem to vary among the faces. Regarding the bilayer structures, the underlying geometry appears to have an impact on dynamical heterogeneity in layer 1 at low temperature or layer 2 at high temperature. The structure of primary faces includes a clear separation of crystalline and defects, whereas the basal face's interface is more disorganized. It is proposed that the basal is more liquid-like and homogeneous at low temperatures. Upon heating to 259 K, large parts of the basal face in Layer 2 melt and it mitigates the confinement effect. The tendency of earlier melting in basal faces was also suggested in experiments by Sanchez et al [18]. One of the characteristics of the secondary face is the frequent switching between Layers 1 and 2. Although the interlayer properties of the secondary prism face do not significantly vary from the bilayer structures, the secondary prism surface exhibits differences in some points. For instance, Layer 3 is more fluidic than in the other faces. In Layer 1, the high degree of long ranged-ordering of liquid-like molecules would be related to the surface roughness.” (page 13)

REVIEWERS' COMMENTS:

Reviewer #1 (Remarks to the Author):

The authors have carefully addressed my major concerns from the previous round of reviews.

I think that the paper is now technically sound. In general, I find the rebuttal to the former reviews satisfactory. I still have a couple of comments:

1. Line 65: King should read Kling
2. Line 397: Does it make sense to discuss these results in the context of astrochemistry? In space, temperatures are normally much lower than -20 C. Even comets should be around 180 K.
3. Line 494: The public unavailability of the $g(x)$ code and the "availability upon request" of data violates the FAIR (Findability, Accessibility, Interoperability, and Reuse) of digital assets principles.

Reviewer #3 (Remarks to the Author):

This is a careful study and the response to all the referees were thorough and well-considered. Now that the rationale is somewhat clearer for using ML and the results have been demonstrated to be invariant to the sampling timescale, I am satisfied that the work is worthy of publication in *commschem* in its current state.

REVIEWERS' COMMENTS:

Reviewer #1 (Remarks to the Author):

The authors have carefully addressed my major concerns from the previous round of reviews.

I think that the paper is now technically sound. In general, I find the rebuttal to the former reviews satisfactory. I still have a couple of comments:

We thank the referee for their insightful comments in the former review.

1. Line 65: King should read Kling

We thank the referee for this correction, and we have fixed this typo.

2. Line 397: Does it make sense to discuss these results in the context of astrochemistry? In space, temperatures are normally much lower than -20 C. Even comets should be around 180 K.

We guess that the few nanoscale porous would be a catalytic site for ice-mediated chemical reactions. For instance, Ice amorphous is formed on interstellar mediators at 20 K and comets at 180 K, and ice is involved in abundant chemical reactions.

We have added sentences,

“In connection with astrochemistry, ice exists in the installer medium and comets at low temperatures (<100 K), and catalytically works for chemical reactions [61]. The surface defects demonstrated at low temperatures (Fig.6c) possibly function as prores to host the chemical reactions. In connection with astrochemistry, ice exists in the installer medium and comets at low temperatures (<100 K), and catalytically works for chemical reactions [60]. The surface defects demonstrated at low temperatures (Fig.6c) possibly function as prores to host the chemical reactions” (page 15)

3. Line 494: The public unavailability of the g(x) code and the "availability upon request" of data violates the FAIR (Findability, Accessibility, Interoperability, and Reuse) of digital assets principles.

We understand this concern from the referee. Since the code to calculate g(x) is covered by the patent (Patent applicant: Keio University. Inventors: K. Yasuoka, D. Yuhara, K. Endo, and K. Tomobe. Application number: JP.2019048988.A. Status of application: published unexamined patent application), we cannot make it publicly available. We decided on its availability upon request, and the pseudo-code is available in Supplementary Algorithm 1

Reviewer #3 (Remarks to the Author):

This is a careful study and the response to all the referees were thorough and well-considered. Now that the rationale is somewhat clearer for using ML and the results have been demonstrated to be invariant to the sampling timescale, I am satisfied that the work is worthy of publication in commschem in its current state.

We thank the reviewer for their thoughtful comments and kind remarks on the revised manuscript.